# Hydrochemical Anomalies in the Vicinity of the Abandoned Molybdenum Ores Processing Tailings in a Permafrost Region (Shahtama, Transbaikal Region)

Nataliya Yurkevich [1,*], Vladimir Olenchenko [1], Andrei Kartoziia [1,2], Tatyana Korneeva [1], Svetlana Bortnikova [1], Olga Saeva [1], Kristina Tulisova [1] and Natalya Abrosimova [1]

[1] Trofimuk Institute of Petroleum Geology and Geophysics, Siberian Branch of the Russian Academy of Sciences, 3 Koptyug Avenue, 630090 Novosibirsk, Russia

[2] Sobolev Institute of Geology and Mineralogy SB RAS, 630090 Novosibirsk, Russia

\* Correspondence: yurkevichnv@ipgg.sbras.ru

**Abstract:** The mobility of chemical elements during the transition from molybdenum ore processing waste to aqueous solutions and the hydrochemical anomalies of a number of elements in surface and underground waters in the vicinity of an abandoned tailings dump were investigated. It is shown that alkaline and alkaline earth metals have high mobility—the main rock-forming components (sodium, lithium, magnesium, strontium), which are released into solution due to leaching from the minerals of the host rocks, as well as metals with zinc, cadmium, manganese, and nickel, which are released into solution due to the dissolution of ore sulfides. Elements with high mobility include Sb, Co, Cu, Be, Se, and Tl. Medium mobility has As, an element of the first hazard class, as well as Mo, Fe, and Pb. Hydrochemical anomalies of cadmium, arsenic, molybdenum, and lead have been determined. The nature of the arsenic and molybdenum anomalies is most likely related to the regional background, while the source of cadmium and lead is most likely the waste studied. The main chemical forms of the presence of elements in the solution of ponds on the surface of tailings ponds are free-ion and sulfate complexes. For example, in the samples of the Shakhtama River and groundwater, we found carbonate, bicarbonate, and hydroxide complexes. The information obtained should be taken into account when planning measures for the purification of surface and groundwater from metals. Additional studies should consider using groundwater in the vicinity of the tailings for drinking water supply.

**Keywords:** tailings; Transbaikal region; Russia; permafrost; neutral mine drainage; metals; metalloids; mobility

## 1. Introduction

Abandoned mine tailings are sources of a wide spectrum of rock-forming, ore and toxic chemical elements which can dissolve from the solid matter due to oxidation by the water and air oxygen. Mine drainage forms as a result of these processes, which has been one of the most widely debated issues in the field of environmental geochemistry within the last 50 years [1–3]. Geochemical and mineralogical reactions, in addition to water, air, and heat transport, are all of key importance in determining waste-rock weathering rates, and thus drainage quality, as well as quantity. Studying the mobility of mine waste pollutants and the underlying mechanisms from the micro- to the macroscale remains important, as expanding mining operations around the world pose increasing potential environmental risks: forward-thinking in the design of long-term dumps that could pose potential leachate quality problems is critical. In recent years, neutral drainage studies have been gaining popularity, in which a wide range of chemical elements that are mobile in neutral and subalkaline conditions are transferred [4–7]. An improved and quantitative understanding of the factors controlling mine waste-rock drainage dynamics, paired with

a growing ability to map in situ heterogeneity in full-scale systems and the ability to utilize large-scale, high-resolution data in practical models, will allow engineers and practitioners to develop more robust prediction models and sustainable management decisions. Continuing research can facilitate optimized waste management and thereby prevent waste-rock-related environmental deterioration in the future.

Exothermic sulfide oxidation (e.g., 1000–1500 kJ·mol$^{-1}$ for pyrite) can cause internal temperatures in waste-rock piles to rise to tens of degrees above average ambient temperatures, e.g., up to 65 °C at some sites [8]. Convection is the combination of conduction (heat diffusion) and advection (bulk fluid flow); radiative transfer and viscous dissipation are typically assumed negligible in waste rock [9,10]. Natural convection arises from buoyancy forces in waste-rock piles generated by oxidation reactions that produce heat and alter the density of the pore gas (e.g., through water vapor) [11]. While the geochemical reactions in weathering waste rock are universal on a molecular level, there is typically large variability in the weathering conditions, waste-rock composition, and drainage composition [12–15]. In this work, emphasis is placed on the mobility of elements as a result of chemical reactions related to rock breakdown by frost action (hypercryogenesis), as when in cold conditions processes are activated along the cracks and the removal of metals and metalloids is intensified [16–19], whereas sulfate and chloride anions, metals, and metalloids acquire mobility [20,21]. Thus, when studying the waste of the Darasun ore node, we showed that cyanide ions and a whole spectrum of ore metals, metalloids, and impurity chemical elements, including arsenic, are mobile [22]. Despite the fact that quite a lot of waste-storage facilities in the world are located in extremely cold conditions (Norway, Canada), the processes of hypercryogenesis are not sufficiently described. This study aims to address the specific gap in geochemistry with regard to the abandoned mine tailings located in extremely cold conditions. In particular, we investigated aspects concerning the mobility of various chemical elements during the transition from waste to solution and the composition of the emerging drainage. The purpose of this work was to show that waste from processing molybdenum ores are sources of a wide range of chemical elements. The objectives of the study were to assess the composition of waste, drainage solutions, river, and groundwater-receiving drainage. The research questions are also intended to determine hydrochemical anomalies and the forms of metals and metalloids in the vicinity of the molybdenum-ore-processing waste storage. The sampling and analyzing of the tailings and water, interacting with solid matter (ponds on the surface of mine tailings dump) for a wide spectrum of chemical elements, allowed us to judge the mobility of these elements due to the oxidation of the solid matter. We have identified the elements with very high mobility, including metals Zn, Cd, Mn, Ni, entering the solution due to the dissolution of sulfides present in the mine tailings. An important finding is that the high mobility is possessed by Sb, Co, Cu, Be, Se, and Tl—the elements that enter the solution due to the dissolution of ore minerals and their impurities, as well as desorption. Much attention should be paid to the behavior of thallium and beryllium, elements of the first hazard class, in our next studies in this area. Additionally, noteworthy is the mobility of arsenic and Mo, Fe, Pb. The sampling and analyzing of the surface and ground waters in the vicinity of the Shahtama tailings for the wide spectrum of chemical elements allowed us to compare the composition of the waters which are affected by drainage from the tailings with the average concentrations in the river waters (clarks). It should be noted that concentrations of a wide range of chemical elements in aqueous solutions in the vicinity of the tailings exceed clark values, including Zn, Cd, Mn, Ni, Sb, Cu, Be, Se, As, Tl, Mo, and Pb, that is, elements characterized by high and very high mobility during the oxidative leaching of elements from waste due to hypercryogenesis. We suppose that hydrochemical anomalies of cadmium and lead in surface and underground waters in the vicinity of the tailings can be associated with the release from the tailings, and the anomalies of molybdenum and arsenic are more likely due to the regional background. The main chemical forms in which chemical elements migrate with aqueous solutions are free-ions and sulfate complexes.

## 2. Materials and Methods

### 2.1. Object

The tailings dump of the Vershino-Shakhtaminsky ore processing plant is located on the outskirts of the village of Vershino-Shakhtaminsky in the Shelopuginsky district of the Transbaikal territory (Figure 1). The climate is characterized as either sharply continental or subarctic according to the Koppen–Geiger classification (Dfc) with long frosty winters but little snow and short warm summers. The average long-term temperature of the coldest month (January) is $-33.6\,°C$ (absolute minimum, $-59\,°C$), the temperature of the warmest month (July)—$17.7\,°C$ (absolute maximum, $36\,°C$). The frost-free period lasts for 74 days. The average annual rainfall is 359 mm.

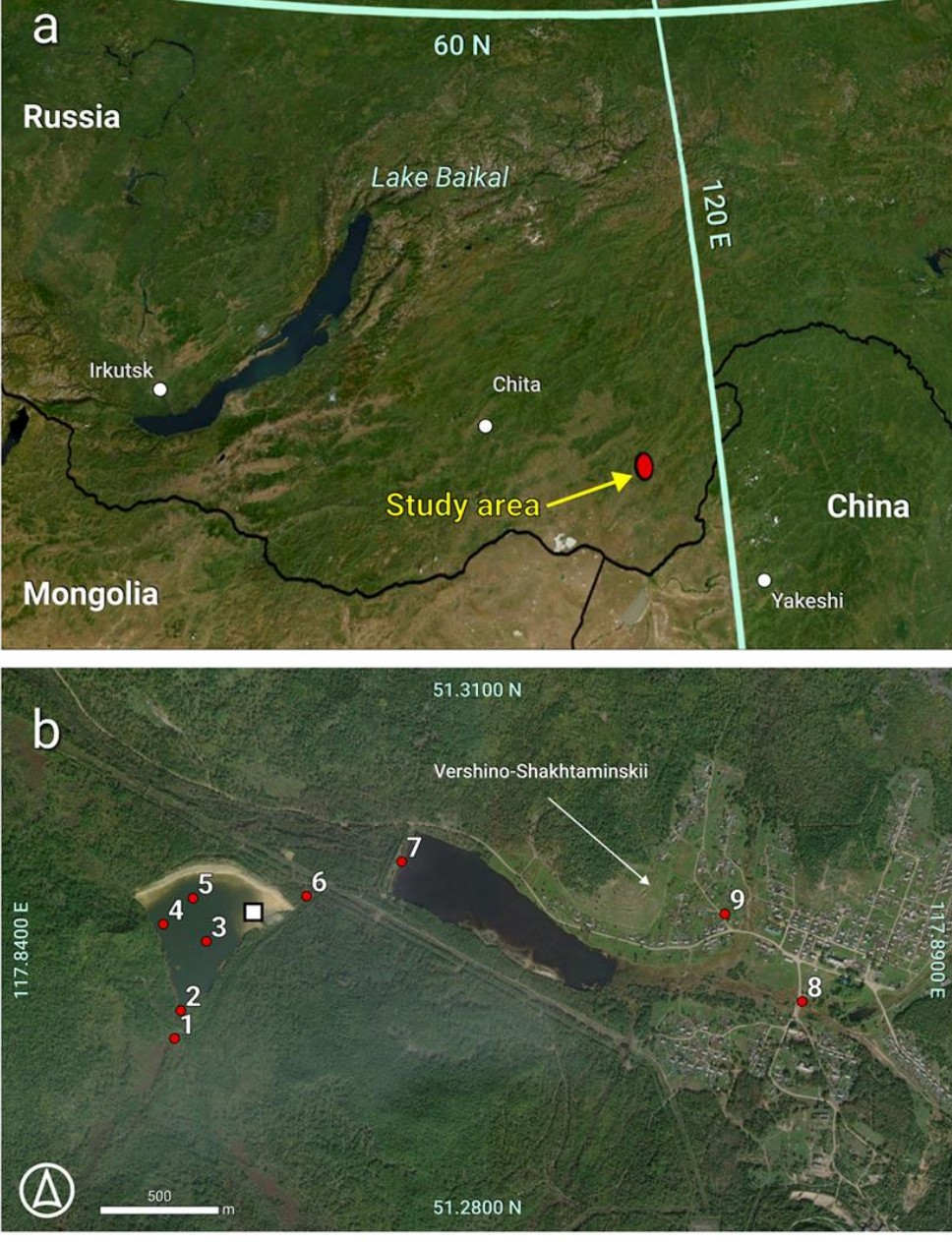

**Figure 1.** Study area: Vershino-Shahtaminskii in the Transbaikal region, Russia (**a**), tailings of the Shahtama molybdenium ores processing (**b**) (points 1–6), storage reservoir (7), Shahtama River (8), drinking water well (9), pit (white square).

The tailings dump is a hydraulic engineering structure with waste flotation, arranged in the intermountain decay by the construction of a dam. The rocks at the base of the tailing dump are of coarse-grained granites, and the waterproofing measures of the base are not provided. The enrichment waste was stored in the period from 1941 to 1993. Currently, the facility is out of ores, and various options for recycling tailings are being considered. Flotation tails are represented by finely dispersed materials with a grain size of −200 mesh (particle size content equal to or less than 0.074 mm is more than 50%). It is a product of grinding copper–molybdenum ores. The main ore minerals are pyrite, molybdenum, galena, sphalerite, and chalcopyrite (Supplementary Materials, Figure S1). Ore-bearing rocks are granites of the Shakhtaminsky complex. In addition to fine fractionation, man-made sediments contain sands from dusty to a medium grain size. For a long time, the central part of the tailings dump was under a layer of water up to 1.5 m deep, which prevented oxygen from entering the section. Along the periphery of the tailings dump, the surface remained dry, which contributed to the active transfer of material during wind erosion, as well as deep seasonal freezing, which resulted in the formation of frost-breaking cracks. In the areas adjacent to the dam, aeolian-relief forms were formed as blow-holes or ripple signs (Figure 2b). Watered soils and their thinly dispersed compositions created favorable conditions for the processes of frost heaving. Cracked mounds of seasonal heaving of up to 0.5 m high were marked along the contour of the flooded part of the tailings dump. At the time of the research, all the water from the central part of the tailings dump was drained through a drainage well (Figure 2a). The surface of the tailings dump has been broken by frost-breaking cracks and drying cracks with a polygonal structure. Frost-breaking cracks are filled with sandy material and form ground veins (pseudomorphoses), which are dissociated by a system of open drying cracks (Figure 2c).

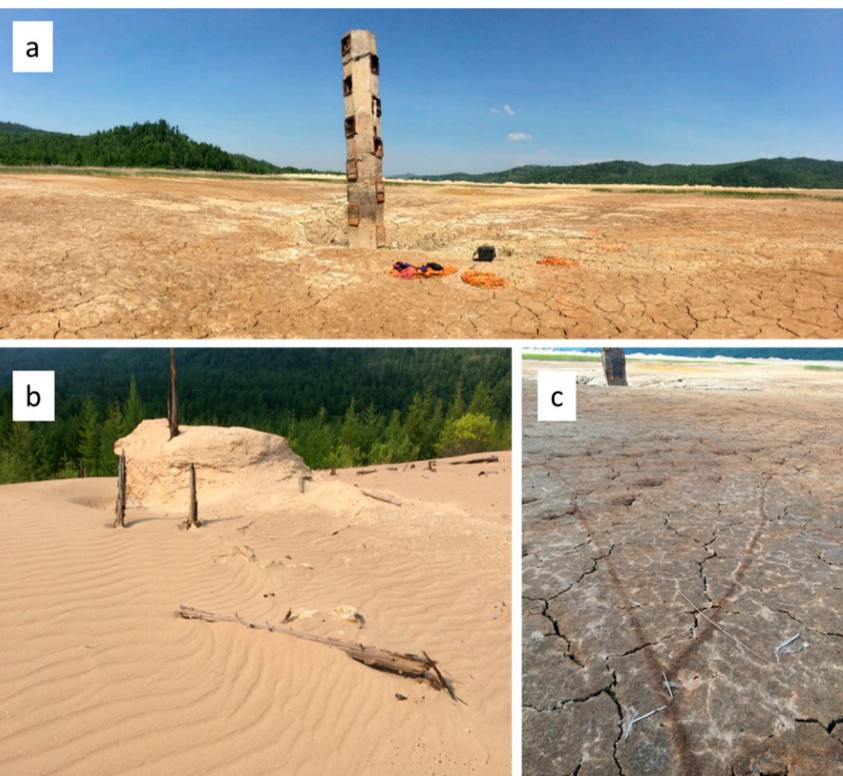

**Figure 2.** Drainage well on the area of the tailings dump (**a**); aeolian processes at the tailings dam (**b**); shrinkage cracks and pseudomorphoses by frost-breaking cracks on the surface of the tailings dump (**c**).

*2.2. Methods*

2.2.1. Sampling and Field Study

During the field and laboratory work, the Shahtama mine tailings area was investigated using geochemical and hydrochemical sampling.

To determine the composition of the solid matter, the Shahtama tailings were sampled at a 120 cm deep pit (white square, Figure 1). In total, eight 200 g samples were collected using a plastic scoop and transferred into polyethylene bags in the laboratory for chemical analysis.

Eight 1 L water samples were collected at each sampling point (Figure 2) at a depth of ≈10 cm. Sample sites included the following: (1) a stream entering the surface of the tailings dump from the decay near the tailings; (2–5) ponds on the surface of the tailings; (6) a natural stream flowing near the tailing dumps at a point lower in the relief; (7) storage reservoir; (8) Shahtama River; (9) drinking water well.

The pH and redox potential (Eh) of water samples were measured in situ using a pH/°C meter (HI 9025 C, Hanna Instruments, Italy) with a combination glass pH electrode (HI 1230 B, Hanna Instruments) and a platinum electrode for the determination of Eh (redox electrode, Hanna Instruments) [23]. Eh's probe was standardized using a Zobell's solution and Ag/AgCl electrode (+228 mV at 25 °C). Electrical conductivity (EC) was determined using a Cond 315 i electrical conductivity meter (WTW, Weilheim, Germany).

Solid samples were mixed with distilled water at a water/solid ratio of 1:2 by weight to prepare pastes according to [24]. The pH, redox potential (Eh), and EC were measured in situ.

The water samples were divided into two series. The first 50 mL aliquot was filtered at the sampling point to <0.25 μm using a microfiltration hydrophobic membrane (Vladipor, Russia) and acidified with distilled $HNO_3$ (99% purity) to reach pH < 2. The second aliquot intended for anion analysis was not filtered. Samples were delivered to the laboratory and stored at <4 °C.

2.2.2. Laboratory Analysis

Solid samples were dried at room temperature for 48 h, homogenized by stirring, sieved using a nylon filter with a pore size of 250 μm (Fritsch, Idar-Oberstein, Villafranca Padovana, Germany), and ground to a particle size of <74 μm by trituration in an agate mortar for volumetric analysis. Their elemental composition was determined using ICP-MS (ELAN-9000 DRC-e, PerkinElmer Instruments LLC, Wellesley, MA, USA) at the Plasma Analytical Center (Tomsk, Russia). High purity argon (99.95%) was used as the plasma-forming, transport, and cooling gas. For tuning, we used a 2% solution of 7Li, 59Co, 89Y, and 205Tl with a concentration of each element being determined at 1 g/L (Tuning Solution, Houston, TX, USA). All measurements were carried out in triplicate (*n* = 3) for each element. The relative standard deviation in all measurements did not exceed 13%. The relative analysis error was 3–8%.

Powder X-ray diffractometry (XRD) was used to determine the phase compositions of crystalline substances and their quantitative phase ratios. X-ray phase studies were performed on an ARL X'TRA powder diffractometer (Thermo Fisher Scientific, Ecublens, SARL, Switzerland) using CuKα radiation, voltage 40 kV, and a current of 25 mA. The diffraction patterns were scanned in the 2θ range from 2° to 65° with a step of 0.02°; the analysis rate was 4° per minute. The morphology and composition of individual grains were studied on a MIRA3 LMU scanning electron microscope (TESCAN, Kohoutovice, Czech Republic) with an INCA Energy 450+ microanalyzer based on an NanoAnalysis X-MAX 80 system (Oxford Instruments, High Wycombe, UK) in the X laboratory. S. L. Soboleva, Novosibirsk (analyst N. S. Karmanov).

Analyses of major cations (Al, Fe, Ca, Mg, K, Na, and Si) and microelements in water samples were carried out using ICP-MS (ELAN-9000 DRC-e, PerkinElmer Instruments LLC, Wellesley, MA, USA) at the Plasma Analytical Center (Tomsk, Russia). The relative error of the analysis was estimated at 2–5% at the mg/L concentration level and 4–8% at the μg/L concentration level.

Another aliquot was analyzed for major anions ($SO_4^{2-}$, $Cl^-$, $HCO_3^-$) using potentiometric, photometric, and titrimetric methods.

The concentration of the sulfate ion was identified using a turbidimetric method. Accuracy and precision were rated as 7% or better at mg/L concentrations.

The concentration of bicarbonate ions in the samples was determined by the titrimetric method. Accuracy and precision were rated at 10% or better at the mg/L concentration level.

The concentration of chlorides in water samples was determined by the titrimetric method with silver nitrate by the formation of insoluble silver chloride. Accuracy and precision were rated as 10% or better at the mg/L concentration level. We additionally tested concentrations of chlorides using ion chromatography (L-Ion 16, Silab, Shanghai, China) and capillary electrophoresis (Kapel-105M, Lumex, Sankt-Petersburg, Russia). Accuracy and precision were rated as 3 and 5% or better, respectively, at the mg/L concentration level.

Species of the elements in the solution and the ability of water to remove them from the solution to form independent solid phases were considered using the software packages Visual Minteq 3.1 [25] and WateQ4F [26].

## 3. Results

### 3.1. The Composition of the Solid Matter

The pH values of the pastes were prepared by mixing the tailings with distilled water at a water/solid ratio of 1:2 by weight range from 2.65 to 6.11 units, and the redox potential (ORP) is from 784 to 530 mV. The lowest pH values and the highest ORP are found at a depth of 54 cm from the surface at the permafrost boundary.

The main minerals of the light fraction in the composition of the waste matter are trioctahedral mica, quartz, kfs, plagioclase, kaolinite, chlorite, smectite, and amphibole. Among the minerals of the heavy fraction are pyrite, barite, goethite, and siderite. Figure 3 shows a fragment of the waste substance (horizon 44–54 cm) from the pit. This sample corresponds to the pH value of the paste as 2.65, and the redox potential as 784 mV. It can be seen that ice is present between the clay layers. Along the cracks—red smudges—are newly formed iron hydroxides.

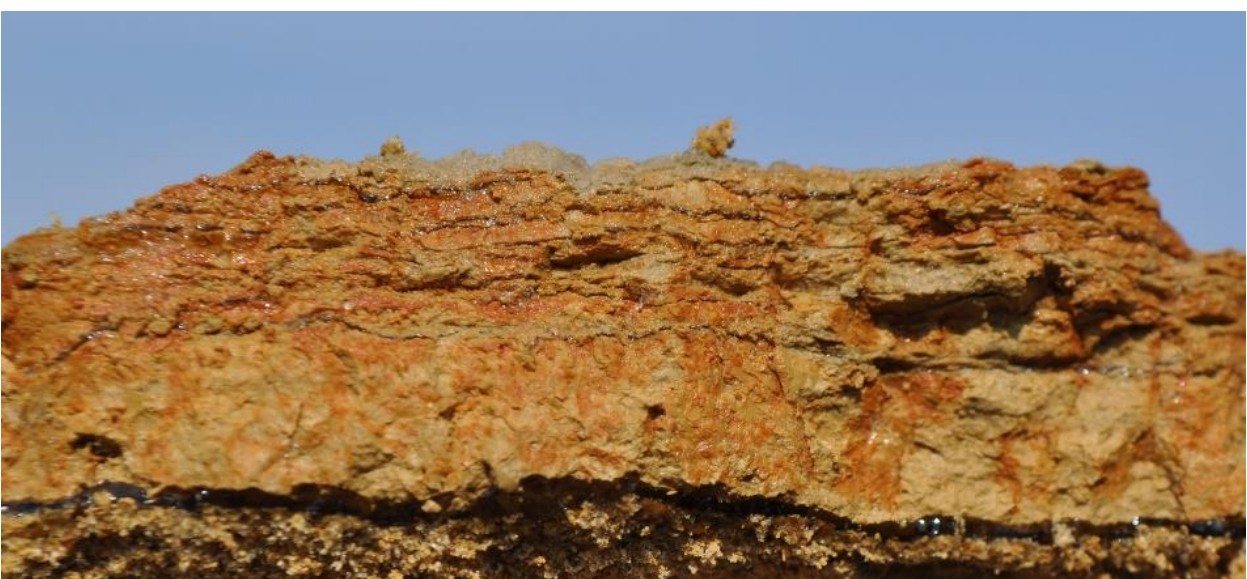

**Figure 3.** A fragment of the substance of Shakhtama tailings from a pit with a depth of 44–54 cm with ice inclusions.

A wide range of chemical elements was determined in the composition of the tailings of the Shakhtama tailing (Table 1): rock-forming Si, Al, K, Fe, Na, Mg, and trace elements (from Ti to In). The largest amounts are Ti, Mn, Ba, Pb, Mo, Cu, Rb, and Sr. It is noteworthy that the tailings under consideration are sources of precious metals: they contain on average

0.51 g/t of gold and 7.4 g/t of silver. We also note the presence of extremely scarce indium (0.42 g/t) and germanium (2.4 g/t). The composition of the waste also includes elements of the first hazard class: Tl (2.7 g/t) and As (65 g/t).

**Table 1.** The element composition of the Shahtama tailings.

| Element | Min | Max | Average |
|---|---|---|---|
| | **%** | | |
| Si | 25 | 30 | 27 |
| Al | 5.4 | 8.9 | 7.3 |
| K | 3.5 | 5.1 | 4.2 |
| Fe | 2.3 | 4.5 | 3.3 |
| Na | 0.46 | 2.8 | 1.7 |
| Mg | 0.29 | 1.6 | 0.8 |
| | **g/t** | | |
| Ti | 2000 | 5200 | 3400 |
| Mn | 210 | 3200 | 1500 |
| Ba | 740 | 2200 | 1400 |
| Pb | 62 | 4000 | 1100 |
| Zn | 130 | 1800 | 880 |
| Mo | 230 | 1000 | 630 |
| Cu | 100 | 1400 | 410 |
| Rb | 190 | 340 | 270 |
| Sr | 87 | 340 | 240 |
| Cr | 18 | 150 | 66 |
| As | 26 | 160 | 65 |
| V | 33 | 76 | 53 |
| Sb | 8.1 | 240 | 48 |
| Li | 20 | 55 | 40 |
| Ni | 21 | 41 | 31 |
| Ga | 14 | 25 | 20 |
| Nb | 12 | 24 | 17 |
| Y | 7.6 | 26 | 16 |
| Th | 6.0 | 22 | 13 |
| Cs | 6.6 | 18 | 11 |
| Sn | 3.1 | 15 | 10 |
| Ag | 0.48 | 24 | 7.4 |
| U | 1.5 | 17 | 6.6 |
| Co | 1.0 | 11 | 5.2 |
| Cd | 0.45 | 11 | 4.6 |
| Tl | 0.88 | 4.1 | 2.7 |
| Ge | 1.8 | 2.7 | 2.4 |
| Te | 1.1 | 3.5 | 1.9 |
| Au | 0.16 | 1.0 | 0.51 |
| In | 0.16 | 1.0 | 0.42 |

*3.2. The Composition of the Water Samples*

The pH value of the water in the stream that falls on the surface of the dump is 6.93 units, its ORP is 498 mV, and the electrical conductivity is 72 μSm/cm (Table 2). In the ponds on the surface of the dump (points 2–4), the electrical conductivity is slightly higher than in the incoming stream. At point 5, the highest ORP and the lowest pH values (5.66 units) are noted, and the electrical conductivity is 320 μSm/cm, while the concentration of the sulfate ion in this solution is 150 mg/L. The pH value of the natural stream flowing near the storage (point 6) is 6.8 units, the electrical conductivity is 65 μSm/cm, which is on the same level as the stream that enters the storage higher in relief (point 1), and the sulfate ion concentration here is 15 mg/L. Note that the electrical conductivity of water in the reservoir is 280 μSm/cm and the pH value is 7.93 units at approximately

the same level of pH and electrical conductivity in the river, and drinking well (Table 2, samples 8, 9).

A wide range of chemical elements was determined in the microelement composition of the studied water (Table 3).

**Table 2.** Physicochemical composition of the water samples: a stream entering the surface of the tailings dump from the decay near the tailings (1); ponds on the surface of the tailings (2–5); natural stream flowing near the tailing dumps at a point lower in the relief (6); storage reservoir (7); Shahtama River (8); drinking water well (9).

| Sample | pH | Eh, mV | X, µSm/cm | mg/L | | |
| | | | | $SO_4^{2-}$ | $Cl^-$ | $HCO_3^-$ |
|---|---|---|---|---|---|---|
| 1 | 6.93 | 498 | 72 | 15 | 0.07 | 40 |
| 2 | 6.80 | 506 | 102 | 22 | 0.08 | 55 |
| 3 | 7.09 | 507 | 74 | 20 | 0.05 | 34 |
| 4 | 6.76 | 512 | 97 | 35 | 0.06 | 23 |
| 5 | 5.66 | 560 | 320 | 150 | 0.05 | 12 |
| 6 | 6.80 | 523 | 65 | 15 | 0.09 | 37 |
| 7 | 7.93 | 489 | 280 | 66 | 1.5 | 92 |
| 8 | 7.42 | 482 | 390 | 110 | 1.7 | 120 |
| 9 | 8.06 | 468 | 320 | 60 | 1.2 | 130 |

**Table 3.** Microelement composition of the surface waters in the surrounding area of the Shahtama tailings (samples 1–9): a stream entering the surface of the tailings dump from the decay (1); ponds on the surface of the tailings (2–5); natural stream flowing near the tailing dumps at a point lower in the relief (6); storage reservoir (7); Shahtama River (8); drinking water well (9). Bold indicates concentrations exceeding clark values for river waters according to [27].

| Element | Sample | | | | | | | | | Clark [1] | MPC WHO [2] | MPC RF [3] |
|---|---|---|---|---|---|---|---|---|---|---|---|---|
| | 1 | 2 | 3 | 4 | 5 | 6 | 7 | 8 | 9 | | | |
| | **mg/L** | | | | | | | | | | | |
| Ca | 5.7 | 9.0 | 5.5 | 7.6 | **33** | 4.7 | 28 | 43 | 36 | 12 | 75 | 180 |
| Mg | 1.9 | 3.0 | 1.9 | 2.5 | **7.3** | 1.7 | 7.5 | 12 | 8.9 | 2.9 | - | 40 |
| K | 1.0 | 1.1 | 1.0 | 1.2 | **4.0** | 1.1 | 2.2 | 2.8 | 2.1 | 2 | - | 50 |
| Na | 4.7 | 5.1 | 5.0 | 6.2 | **10** | 6 | 10 | 12 | 7.3 | 5 | 200 | 120 |
| Si | 4.5 | 4.4 | 2.7 | 3.2 | 2.9 | 4.4 | 0.2 | 4.2 | 8.1 | 6 | - | - |
| | **µg/L** | | | | | | | | | | | |
| Al | 57 | 21 | 16 | 30 | **240** | 38 | 37 | 31 | 27 | 160 | 200 | 40 |
| Fe | 126 | 38 | 46 | 103 | 32 | 49 | 33 | 66 | **408** | 40 | 300 | 100 |
| Sr | 45 | 77 | 45 | 59 | **216** | 39 | **336** | 460 | 487 | 30 | - | 400 |
| Ba | 7.4 | 9.2 | 8.7 | 10 | **22** | 7.3 | 12 | 19 | 10 | 20 | 700 | 700 |
| Ni | 1.4 | 1.5 | 0.86 | 1.7 | **10** | 1.5 | 1.9 | 2.8 | 1.5 | 20 | 70 | 20 |
| Mn | 14 | 12 | 9.5 | **410** | **2800** | 9.2 | **20** | **29** | **160** | 3.0 | 400 | 100 |
| Te | 0.008 | 0.005 | 0.007 | 0.005 | 0.013 | 0.008 | 0.004 | 0.008 | 0.005 | 3.0 | - | 3 |
| Rb | 1.4 | 1.7 | 1.9 | **3.0** | **13** | 1.7 | **4.8** | **6.8** | **6.2** | 2.6 | - | 100 |
| Li | 1.6 | 2.3 | 2.0 | **2.9** | **9.5** | 1.5 | **3.1** | **4.3** | **6.6** | 2.5 | - | 80 |
| Y | 0.2 | 0.2 | 0.1 | 0.3 | 0.2 | 0.2 | 0.1 | 0.1 | 0.2 | 2.5 | - | - |
| Nd | 0.2 | 0.2 | 0.1 | 0.3 | 0.1 | 0.2 | 0.0 | 0.1 | 0.1 | 2.5 | - | - |
| W | 0.03 | 0.05 | 0.06 | 0.05 | 0.02 | 0.02 | 0.17 | 0.06 | 2.1 | 2.0 | - | 0.8 |
| Ti | **1.5** | **1.1** | 0.6 | 0.9 | 0.7 | **1.4** | 0.2 | 1.1 | **1.5** | 1.0 | - | 60 |
| Cr | 0.62 | 0.45 | 0.35 | 0.28 | 0.13 | 0.48 | 0.22 | 0.33 | 0.27 | 1.0 | 50 | 20 |
| Zn | **13** | **11** | **25** | 190 | 3300 | **11** | 7 | **24** | **8** | 1.0 | 3000 | 1000 |
| Sb | 0.40 | 0.97 | **1.3** | **2.2** | **4.6** | 0.41 | **5.9** | **4.4** | **1.1** | 1.0 | 20 | 5 |

**Table 3.** *Cont.*

| Element | Sample | | | | | | | | | Clark [1] | MPC WHO [2] | MPC RF [3] |
|---|---|---|---|---|---|---|---|---|---|---|---|---|
| | 1 | 2 | 3 | 4 | 5 | 6 | 7 | 8 | 9 | | | |
| | mg/L | | | | | | | | | | | |
| Ce | 0.37 | 0.22 | 0.07 | 0.50 | 0.27 | 0.26 | 0.11 | 0.13 | 0.17 | 1.0 | - | - |
| Lu | 0.01 | 0.01 | 0.01 | 0.01 | 0.01 | 0.01 | 0.01 | 0.01 | 0.01 | 1.0 | - | - |
| Co | 0.10 | 0.09 | 0.07 | 0.12 | 0.44 | 0.08 | 0.16 | 0.21 | 0.27 | 0.7 | n.d. | 100 |
| Sc | 0.10 | 0.10 | 0.10 | 0.10 | 0.10 | 0.10 | 0.10 | 0.10 | 0.10 | 0.30 | - | - |
| Se | 0.05 | 0.16 | **0.35** | 0.15 | **0.42** | 0.03 | 0.03 | 0.20 | 0.02 | 0.20 | 40 | 2 |
| Ag | 0.01 | 0.01 | 0.02 | 0.06 | 0.02 | 0.02 | 0.01 | 0.01 | 0.01 | 0.20 | 50 | 50 |
| In | 0.01 | 0.01 | 0.01 | 0.01 | 0.01 | 0.01 | 0.01 | 0.01 | 0.01 | 0.20 | - | - |
| Th | 0.13 | 0.03 | 0.02 | 0.03 | 0.02 | 0.09 | 0.00 | 0.01 | 0.02 | 0.10 | - | - |
| U | **0.45** | **0.30** | **0.17** | **0.17** | 0.06 | **0.29** | 15 | 28 | **4.0** | 0.10 | - | - |
| Cu | **5.5** | **7.2** | **13** | **29** | **34** | **6.7** | **4.9** | **9.1** | **2.7** | 0.08 | 2000 | 1000 |
| Mo | **1.7** | **10** | **28** | **27** | **4.2** | **1.4** | **96** | **120** | **34** | 0.07 | 70 | 1 |
| La | **0.24** | **0.18** | **0.07** | **0.30** | **0.21** | **0.18** | **0.04** | **0.07** | **0.10** | 0.04 | - | - |
| V | 0.05 | 0.15 | 0.10 | 0.05 | 0.06 | 0.22 | 0.05 | 0.27 | 0.24 | 0.02 | - | 1 |
| Cs | 0.03 | 0.02 | 0.04 | 0.11 | **0.43** | 0.02 | 0.08 | 0.17 | 0.70 | 0.01 | - | - |
| Ga | 0.02 | 0.01 | 0.01 | 0.04 | **0.19** | 0.01 | 0.01 | 0.01 | 0.02 | 0.01 | - | - |
| Be | 0.02 | 0.01 | 0.01 | 0.01 | **0.17** | 0.01 | 0.01 | 0.01 | 0.02 | 0.01 | 12 | 0.2 |
| As | **1.21** | **1.07** | **1.02** | **1.43** | **1.37** | **0.99** | **2.00** | **0.96** | **2.98** | 0.004 | 10 | 10 |
| Sn | **0.12** | **0.10** | **0.06** | **0.10** | **0.08** | **0.05** | **0.05** | **0.08** | **0.06** | 0.004 | 1 | 0.12 |
| Cd | **0.15** | **0.15** | **0.73** | **0.92** | **16** | **0.19** | **0.29** | **0.82** | **0.14** | 0.002 | 3 | 1 |
| Tl | **0.01** | **0.01** | **0.01** | **0.04** | **0.10** | **0.01** | **0.01** | **0.02** | **0.01** | 0.001 | - | - |
| Pb | **4.0** | **2.5** | **1.8** | **10** | **4.0** | **2.0** | **1.8** | **3.7** | **1.7** | 0.001 | 10 | 10 |

[1] Clark is the average concentration in the waters of hydrosphere according to [27]. [2] MPC WHO (maximum permissible concentrations of elements in drinking waters, World Health Organization) [28]. [3] MPC RF (maximum permissible concentrations of elements in water, Russian Federation) [29].

## 4. Discussion

In aqueous solutions, among the elements whose concentrations exceed the clark values for river waters [27] is a wide spectrum of elements: Ca, Mg, K, Na, Al, Sr, Ba, Ni, Mn, Rb, Li, Zn, Sb, Se, Cu, Mo, La, Cs, Ga, Be, As, Sn, Cd, Tl, and Pb.

Concentrations of Al, Mn, Zn, and Cd in the waters of the pond on the surface of the tailings exceed MPC according to the WHO [28] and RF [29] standards (sample 5, Table 3). Concentrations of Mo is higher than MPC RF in almost all studied samples. Its concentration varies in the range from 1.4 µg/L (natural spring, sample 6, Table 3) to 120 µg/L (Shahtama River, Sample 8, Table 3). We provide some comparison with regard to the detected concentrations of Mo in the mining-affected waters, which are described in various literary sources. For example, according to [30], the groundwater at 8.4 m from the tailings of the Ylöjärvi mine of Cu–W–As ore type (Finland) contains 32.3 µg/L of Mo. Groundwater in the vicinity of the historic mines within the San Antonio-El Triunfo district, Mexico (Au–Ag–Pb–Zn–As ore type) contains 5–150 µg/L [31]. The drinking water well in the Vershinno-Shahtaminsky village contains 34 µg/L of Mo. As for river waters, the concentrations of Mo in the river waters within the vicinity of the Gyama Cu-polymetallic plant in Central Tibet is equal to 0.6–9.7 µg/L according to [32] and 1.12–1.28 µg/L in the Kocacay river water in the vicinity of the Balya Pb–Zn–Ag mine (Turkey) [33]. It is noteworthy that the concentrations of Mo in the Shakhtama river is more than 100 times higher than these values.

In order to trace the relationship between the microelement composition of water in surface streams near the storage facility and the composition of potential drainage from solid matter, we firstly considered the features of the composition of water in a pond on the surface of the tailings (sample 5, Table 3). The water here is characterized by low pH values and high Eh, which indirectly indicates oxidation processes during hypercryogenesis. This is also evidenced by the low pH values of the pastes in the pit near this pond (sample 5, Table 2).

In the composition of the sample of this pond, we distinguish five groups of elements whose concentrations exceed the clark values of river waters [27] and at the same time are significantly higher than the concentrations in other studied samples: rock-forming elements, the main components of host rocks, and Ca, Mg, K, Na, Al, Cs, Rb, Li, and Sr; ore metals Mn, Cu, Zn, Cd, and Pb; impurity metals Co, Ni, Ba, Se—a chalcophile element with an affinity for sulfur, similarly released into solutions due to the oxidation of sulfide residues in the tailings; and Be and Tl—elements of the first hazard class, having high mobility in slightly acidic and neutral media [34–36]. The data obtained indirectly indicate that the sources of these elements in surface waters near the storage are the processes of tailings oxidation during the "water–rock" interaction.

As for arsenic, the question of its genesis is highly disputable: its concentration in ponds on the surface of the repository (sample 4 and 5, Table 3) is 2.5 times lower than in samples from the drinking well in the Vershinno-Shakhtaminsky village (sample 9) and 1.5 times lower than in the reservoir (point 7, Table 3) and is at the same level with the concentrations in the natural stream flowing near the reservoir (samples 1 and 6) and Shakhtama River (sample 8). We have not found data on arsenic contamination of surface and underground waters in the vicinity of molybdenum deposits and their tailings dumps; however, there is data on soil contamination [37].

The situation with molybdenum is similar—its concentrations are significantly higher both in the Shakhtama River and in the reservoir in comparison with water samples from ponds on the surface of the reservoir, which indirectly indicates that the presence of both arsenic and molybdenum in surface waters is primarily associated with the regional background of the considered ore region.

To assess the mobility of chemical elements during the transition from waste to an aqueous solution, the mobility coefficients were calculated using the following formula:

$$MC = \frac{C_{water}}{C_{solid}} \cdot 10^5$$

where $C_{water}$—the concentration of the element in solution (sample 5, Table 3), ppm; $C_{solid}$—the concentration of the element in a solid, ppm (average, Table 1); and $10^5$—conversion factor for easy visualization in the form of a diagram (Figure 4).

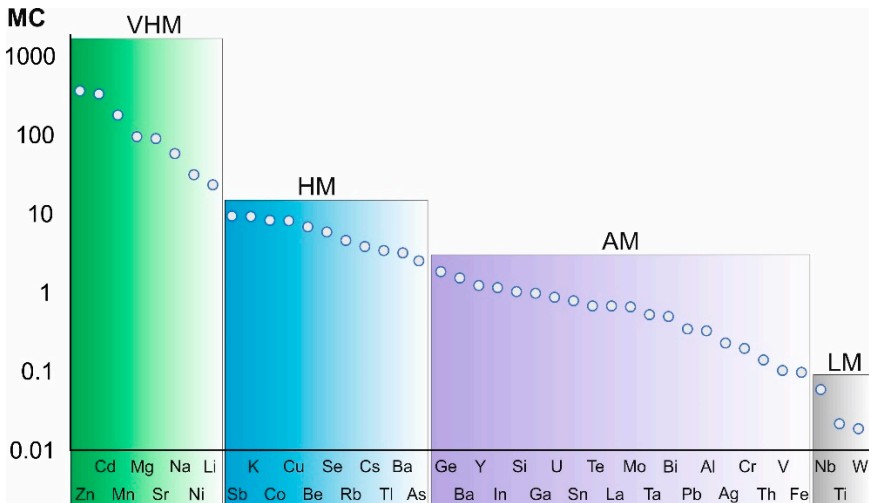

**Figure 4.** Mobility of chemical elements during the transition from Shahtama tailings to solution. MC is the mobility coefficient. VHM, HM, AM, LM—four groups of very high, ultra-high, medium, and low mobility, respectively.

Upon analyzing the mobility of chemical elements in the "solid-solution" system, we distinguish four groups of elements with very high, high, medium, and low mobility (Figure 4). Elements with very high mobility (VHM, MC = 24–368) include: (1) alkali and alkaline earth metals—the main rock-forming components (Na, Li, Mg, Sr) released into solution due to leaching from minerals of host rocks (trioctahedral mica, kfs, plagioclase, kaolinite, chlorite, smectite, amphibole) during congruent and incongruent dissolution and ion exchange and (2) metals Zn, Cd, Mn, and Ni entering the solution due to the dissolution of sulfide minerals ores. Elements with only high mobility (HM, MC = 3–10) include: (1) elements (Sb, Co, Cu, Be, Se, Tl) that enter the solution due to the dissolution and desorption of impurity components. Attention is drawn to beryllium and thallium, elements of the first hazard class; their concentrations in the solution of the pond on the surface of the tailings are equal to 0.17 and 0.1 μg/L (sample 5, Table 3), which is 17 and 100 times higher than the average clark, respectively [27]. (2) K, Cs, Ba, and Rb—elements entering the solution due to the dissolution of the minerals of the host rocks. The large group with medium mobility (AM, MC = 0.1–2.1) includes As, an element of the fist hazard class, Mo, Fe, and Pb. Finally, the inactive group includes (LM, MC = 0.02–0.05) Nb, Ti, and W.

The principal component analysis (PCA) allowed us to identify two main groups of studied water bodies: I—samples 1, 2, 3, 4, 6 (a stream entering the surface of the tailings dump from the decay (1); ponds on the surface of the tailings (2–4); natural stream flowing near the tailing dumps at a point lower in the relief (6)) and II—samples 5, 7, 8, 9 (a pond on the surface of the tailings with low pH values and high Eh, which indirectly indicating oxidation processes during hypercryogenesis (5), storage reservoir (7), Shahtama River (8), drinking water well (9) (Supplementary Materials, Figure S2). These data indirectly indicate that the trace element composition of water in the storage reservoir (7), Shahtama River (8), drinking water well (9) may be due to the migration of chemical elements with water drainage formed during the oxidation of Shakhtama tailings and leaching of elements from them.

Hydrochemical anomalies of representatives of the groups VHM, HM, AM: Cd, As, Mo, and Pb and their chemical forms in aqueous solutions were considered. Figure 5 shows hydrochemical anomalies of cadmium, arsenic, molybdenum and lead, expressed

as the ratio of their concentrations in water samples to clarks in river water [27]. We note significant anomalies of cadmium and lead in ponds on the storage surface (up to 7754 and 9657, respectively, in samples 5 and 4) and in the Shakhtama river (408 and 3688, respectively, sample 9, Figure 5). As for molybdenum and arsenic, we note pronounced anomalies only for water from the reservoir, the Shakhtama River, and the groundwater well (samples 7, 8, 9). As we mentioned above, we believe that anomalies in the surface and underground waters of Mo and As are mainly associated with the regional background of this ore region.

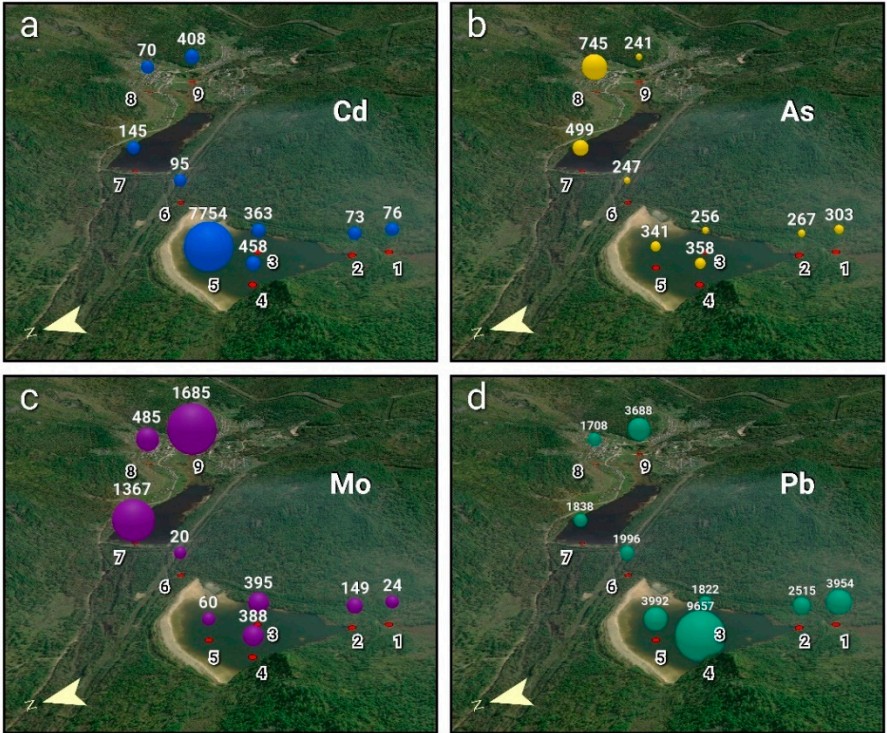

**Figure 5.** Hydrochemical anomalies of Cd (**a**), As (**b**), Mo (**c**), and Pb (**d**) in samples of surface and ground waters in the area of the Shakhtama tailing dump. Numerals indicate the excess clark ratio for river waters [27].

The analytical determination of the total content of elements in waters, carried out in order to assess the impact of man-made factors on them, is currently recognized as insufficient and often biased. The reason is the fact that the migration ability of elements is determined not so much by the total (gross) content, but rather by the ratio of the existing forms of their presence in the studied environment [38,39]. Speciation results by Visual Minteq/Wateq4f suggests that the forms of Mo migration are presented mainly as complexes $MoO_4^{2-}$, $MgMoO_{4(aq)}$, $CaMoO_{4(aq)}$, and an insignificant fraction of $HMoO_4^-$ and $MoO_3(H_2O)_{3(aq)}$. The share of $MoO_4^{2-}$ in the Shakhtama riverbed decreases downstream and amounts to 72–75% of the total share content due to an increase in the $MgMoO_{4(aq)}$ and $CaMoO_{4(aq)}$ aquacomplexes. Cadmium migrates for the most part in all sampling sites in the form of free $Cd^{2+}$ ions, which slightly decreases due to an increase in sulfate and carbonate aqua complexes $CdSO_{4(aq)}$ and $CdCO_3(aq)$. For Pb, the greatest variety of occurrence forms was noted, but $Pb^{2+}$ and $PbCO_{3(aq)}$ were dominant. The proportion of sulfate forms in sample 5 (24%) after a decrease in the technogenic load is replaced by the formation of carbonate complexes $PbCO_{3(aq)}$, $Pb(CO_3)_2^{2-}$, and $PbHCO_3^+$, which is probably associated with a change in pH towards alkalization. Arsenic migration forms are mainly represented by $HAsO_4^{2-}$ and $H_2AsO_4^-$ complexes. When the physicochemical conditions in the well and in the river change, $HAsO_4^{2-}$ is the predominant complex, while in the technogenic pond, mainly arsenic is represented by $H_2AsO_4^-$.

The main mineral phases in relation to which solutions of the pond on the surface of the tailings are supersaturated are Al-hydroxides and hydrosulfates (Table S1). The surface of newly formed aluminum hydroxides has a high sorption capacity. Metals and impurity elements located in the dissolved part of the studied water bodies in the vicinity of Shakhtama mine tailings can be sorbed on the surface of such minerals and co-precipitate with them. Thanks to this process, such water bodies have a high self-cleaning potential.

## 5. Conclusions

1.  The composition of molybdenum ore processing wastes contained in the abandoned Shakhtama storage was determined. The largest amounts are Ti, Mn, Ba, Pb, Mo, Cu, Rb, and Sr. The tailings under consideration contain precious metals: on average 0.51 g/t gold and 7.4 g/t silver. In addition, they contain extremely scarce components, indium (0.42 g/t) and germanium (2.4 g/t). The composition of the waste also includes elements of the first hazard class: Tl (2.7 g/t) and As (65 g/t).

2.  Waste has been shown to be a source of a wide range of chemical elements migrating with water flows. Elements pass into an aqueous solution due to the oxidation and dissolution of ore minerals that remain in the waste. This is also due to hypercryogenesis reactions: weathering of tailings material under the influence of seasonal temperature difference and when the alternation of water freezing and thawing leads to the material cracking and the intensification of its oxidation.

3.  In aqueous solutions in the vicinity of the tailings, among the elements whose concentrations exceed the clark values for river waters, there are Ca, Mg, K, Na, Al, Sr, Rb, Ba, Ni, Mn, Li, Zn, Sb, Se, Cu, Mo, La, Cs, Ga, Be, As, Sn, Cd, Tl, and Pb.

4.  Elements with very high mobility include alkaline and alkaline earth metals, i.e., the main rock-forming components (Na, Li, Mg, Sr), which enter the solution due to leaching from minerals of the host rocks, and metals (Zn, Cd, Mn, Ni), which enter the solution due to the dissolution of sulfides present in the composition of ores. Elements with high mobility include Sb, Co, Cu, Be, Se, and Tl, which enter the solution due to the dissolution of ore minerals and their impurities, as well as desorption. It should be noted that Shakhtama waste is a potential source of beryllium and thallium, elements of the first hazard class. Elements with medium mobility include As, an element of the first hazard class, as well as Mo, Fe, Pb. Elements Nb, Ti, and W are inactive.

5.  Hydrochemical anomalies of cadmium and lead in surface and underground waters in the vicinity of the tailings can be associated with the release of the studied waste from the substance, and the anomalies of molybdenum and arsenic are more likely due to the regional background.

6.  The main chemical forms due to the presence of elements in the solutions of ponds on the surface of the tailings are free-ion and sulfate complexes. In the samples of Shahtama River and ground waters, these include carbonate, hydrocarbonate, and hydroxide complexes.

**Supplementary Materials:** The following supporting information can be downloaded at: https://www.mdpi.com/article/10.3390/w15081476/s1, Figure S1: The XRD patterns for the phase composition of crystalline substances and their quantitative phase ratios; Figure S2: Principal component plot as a result of the principal component analysis for the microelement composition of the surface waters in the surrounding area of the Shahtama tailings (samples 1–9, Table 3); Table S1: Saturation indices of minerals in water samples of a stream entering the surface of the tailings dump from the decay (1), pond on the surface of the tailings (5), Shahtama River (8), drinking water well (9).

**Author Contributions:** Conceptualization, N.Y. and A.K.; methodology, N.Y., A.K., V.O. and O.S.; validation, K.T., N.Y. and O.S.; formal analysis, N.Y., N.A. and O.S.; investigation, N.Y., V.O., A.K., O.S. and K.T.; resources, N.Y. and S.B.; data curation, K.T., N.Y. and T.K.; writing—original draft preparation, N.Y., A.K. and T.K.; writing—review and editing, N.Y.; visualization, A.K., N.Y. and T.K.; supervision, N.Y.; project administration, N.Y.; funding acquisition, N.Y. All authors have read and agreed to the published version of the manuscript.

**Funding:** This research was funded by Ministry of Education and Science of Russian Federation, grant number FWZZ-2022-0029, Russian Foundation for Basic Research, grant number 20-05-00336. Work was conducted on the state assignment of IGM SB RAS (No. 122041400252-1).

**Institutional Review Board Statement:** Not applicable.

**Informed Consent Statement:** Informed consent was obtained from all subjects involved in the study.

**Data Availability Statement:** Data are available upon request to the corresponding author.

**Conflicts of Interest:** The authors declare no conflict of interest.

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
