# Peer review of "Hydrochemical Anomalies in the Vicinity of the Abandoned Molybdenum Ores Processing Tailings in a Permafrost Region (Shahtama, Transbaikal Region)"

_water, doi:10.3390/w15081476_

Round 1

Reviewer 1 Report

Please find the attached review comments.

Author Response

Dear Reviewer, thank you for your careful reading, appreciation of our work and valuable comments. We have tried to take into account all your comments. Below are our answers to each of them. In the text of the article, corrections according to your recommendations are highlighted in green.

Question or comment

Answer

It would be helpful to clarify what specific gap in knowledge this study aims to address and how it advances the field. The introduction mentions the general topic of the study, but it would benefit from more specific information on the research questions, methodology, and findings. This could be done in later sections of the paper, but including some key details in the introduction would help to engage the reader and provide a clearer sense of the study's scope and purpose

Thank you. We tried to clarify what specific gap in knowledge this study aims to address and how it advances the field; to add key details on the research questions, methodology, and findings

L34  "dilute" should be replaced with "dissolve" or "leach" as it is more accurate in describing the process of chemical elements being released from mine tailings into the environment.

The "dilute" was replaced with "dissolve"

L35  "is the on of the discussible question" should be revised to "is one of the debated issues" or "is a topic of discussion."

The "is the on of the discussible question" was replaced by "is one of the debated issues".

L47  "and the power to harness" should be revised to "and the ability to utilize."

The "and the power to harness" was replaced by "and the ability to utilize."

L60  "waste-rock and drainage composition" should be revised to "waste-rock composition and drainage composition" to improve clarity.

The "waste-rock and drainage composition" was replaced by "waste-rock composition and drainage composition"

L74  "the tasks were" could be revised to "the objectives of this study were."

The "the tasks were" were revised to "the objectives of this study were."

The quality of Figure 1 should be improved, especially by adding more detailed coordinates.

We have improved the quality of Figure 1 and have added more detailed coordinates

Figures 2 to 4 should be combined into one image.

We combined Figures 2 to 4 into one image.

Combine subsection Field and Sampling.

Subsections Field and Sampling were combined.

XRD patterns should be provided

We provided the XRD patterns in the supplementary materials (Fig. 1S).

Table Please change the commas between the numbers in the table to periods in all Tables.

In tables, we use commas as decimal separators, that is, signs used to separate the integer and fractional parts of a number. We replaced them with dots.

Authors mentions the chemical elements with high and medium mobility and the hydrochemical anomalies identified, it would be helpful to provide more specific details on the concentrations of these elements and anomalies and how they compare to regulatory or safety standards.

Yes, thank you. We provided the concentrations of the elements in the Table 3. Anomalies of Cd, As, Mo, Pb in comparison with Clark for river waters (according to Taylor, S.R.; McLennan, S.M. The Continental Crust: Its Composition and Evolution; OSTI: Oak Ridge, TN, USA, 1985) are provided on the Fig. 5 of the revised manuscript. The clarks values are provided in the Table 3 in order for each reader to be able to independently compare the concentrations we found with the clarks. Also we added in the Table 3 regulatory standard (Maximum Permissible Concentration, Russian Federation (MPC)) and WHO standards and some text on the comparison of the concentrations of the chemical elements with clarks and MPC.

I do not think the authors have make a very reasonable discussion based on the results. A good discussion should focus on explaining and evaluating what you found, showing how it relates to your literature review, and making an argument in support of your overall conclusion. Although the results are good, but it is better to reorganize this section.

Yes, thank you. We tried to reorganize this section.

It would be better if contour lines of the terrain could be added to Figure 7.

Thank you for the suggestion! However, we guess that lines of the terrain will be not informative and will complicate this figure. We kindly ask reviewer to not change the figure.

L238 "we first of all considered the features of the composition of water", it would be more clear to replace "first of all" with "firstly" or "first".

The "first of all" was replaced by "firstly".

L240  "The water here is characterized by low pH values and high Eh, which indirectly indicates oxidation processes during hypercryogenesis", it would be clearer to replace " which indirectly indicates " with "indicating".

The "which indirectly indicates" was replace with "indicating".

Figure 8 The chemical symbols in Figure 8 do not have superscripts.

Yes, thank you. We tried to change the chemical symbols (Fig. 6 in the revised version).

Reviewer 2 Report

The manuscript water-2282190 presents some results on anomalies in metallic and trace elements present in surface and groundwater. It should be of sure interest to readers of the journal, but some text improvements are necessary. For example, although the term "hypergryogenesis" is mentioned in the work to define the process of disintegration and chemical attack of rocks at low temperatures, there is no climatic description of the study area, nor temperature data of the analyzed waters are provided in Table 1. This is of fundamental importance because without it it is not clear at what temperature the specification analysis was done by thermodynamic codes used by the authors (Visual Minteq/Wateq4F).
It would also be interesting to associate a multivariate statistical analysis as HCA and/or PCA to the mobility groups found by the authors.
Finally, other things that should be added to the job:
- please add to table 1 the values of the cation-anion charge balance for each sample, and also report the other major constituents present in the first part of table 2;
- please indicate in a supplementary file the saturation indexes of the minerals of interest (see comment on figure 6 in the attached file)
Other minor comments are in the attached pdf file.
Hope this helps

Author Response

Dear Reviewer, thank you for your careful reading, appreciation of our work and valuable comments. We have tried to take into account all your comments. Below are our answers to each of them. In the text of the article, corrections according to your recommendations are highlighted in yellow.

Question or comment

Answer

Technical corrections of the text.

Dear Reviewer, thank you for the careful analysis of the manuscript and corrections. We corrected the article according to your comments (all changes marked by yellow).

Please add a subsection describing the climate of the studied area (Koppen-Geiger classification, rainfall, average min max temperature).

Yes, thank you. We added information on the climate of the studied area in the subsection 2.1 (Object).

L. 135. Please specify if the Eh's probe was standardized using a calibration solution (specifying the mV value)

Yes, Eh's probe was standardized using a Zobell’s solution and Ag/AgCl electrode (+228 mV at 25 0С). The information was added in the text.

L. 174. Of what analysis? At line 179 and 182 there other two are accuracy and precision values

Yes, thank you. We added the information in the text.

Table 2. I think that chloride by Mohr method is not useful at this low salinity because the detection limit is 0.15 mg/l. At lower concentrations, ion chromatography would be better.

See "500-Cl A. Introduction" and the following method in Standard Methods: https://www.standardmethods.org/

Baird, R., & Bridgewater, L. (2017). Standard methods for the examination of water and wastewater. 23rd edition. Washington, D.C., American Public Health Association.

Yes, thank you. We tested concentrations using ion chromatography and capillary electrophoresis.

Table 2. I suggest to include the other major consituents in the first part of Table 3, and add the cation-anion charge balance as % of each sample. This value is often included in the output from software as Visual-minteq and/or Wateq4f

Yes, thank you. We added this information in the Table 2.

Table 3. Why not to add a column with WHO guideline values? https://www.who.int/publications/i/item/9789240045064

Yes, thank you. We added the WHO guideline values and MPC for Russian Federation.

L. 284. It would be interesting to compare this element groups within those obtained by cluster analysis (HCA) and/or principal component analysis (PCA). Take a look to the following manuscript:

Ayari J, Barbieri M, Barhoumi A, Boschetti T, Braham A, Dhaha F, Charef A (2023) Trace metal element pollution in media from the abandoned Pb and Zn mine of Lakhouat, Northern Tunisia.

Journal of Geochemical Exploration 247. DOI: 10.1016/j.gexplo.2023.107180

Thank you so much for your recommendation. Unfortunately, we could not get access to the full text of the article to read the material carefully. Nevertheless, we made a PCA for the data from Table 3 and identified two main groups of studied water bodies: I – samples 1, 2, 3, 4, 6 ( a stream entering the surface of the tailings dump from the decay (1); ponds on the surface of the tailings (2-4); natural stream flowing near the tailing dumps at a point lower in the relief (6)) and II – samples 5, 7, 8, 9 (a pond on the surface of the tailings with low pH values and high Eh, which indirectly indicating oxidation processes during hypercryogenesis (5), storage reservoir (7), Shahtama River (8), drinking water well (9) (Fig. 2S, supplementary materials). These data indirectly indicate that the trace element composition of water in the storage reservoir (7), Shahtama River (8), drinking water well (9) may be due to the migration of chemical elements with water drainage formed during the oxidation of waste from the Shakhtaminsky storage and leaching of elements from them. We provide Figure 1S only in supplementary materials, and our comments on its interpretation are only in responses to the reviewer, since we believe that PCA and HCA data analyses deserve separate detailed consideration and publication in the form of our next article. We thank the reviewer for his valuable recommendation on the development of this topic.

L. 303. Speciation results by Visual Minteq/Wateq4f suggests that

Yes, thank you. The phrase was added.

L. 310. what is the results of the mineral saturation indices (S.I.) for these elements? I suggest to add a supplementary file with an excel file with S.I. It should be noted that elements could be adsorbed on the surface of supersatured minerals as sulfates, Al-Fe-Mn-oxides, carbonates. Therefore it would be useful for the reader to see the S.I also for these minerals

Yes, thank you. We added the supplementary file with SI indices (Tab. 1S). According to the calculating saturation indices the main mineral phases in relation to which solutions of the pond on the surface of the tailings are supersaturated are Al-hydroxides and hydrosulfates (Tab. 1S). We agree with the honored reviewer that the surface of newly formed aluminum hydroxides has a high sorption capacity. Metals and impurity elements located in the dissolved part of the studied water bodies in the vicinity of Shakhtama mine tailings can be sorbed on the surface of such minerals and co-precipitate with them. Thanks to this process, such water bodies have a high self-cleaning potential.

Round 2

Reviewer 2 Report

The authors' response and revisions have satisfactorily addressed my comments on the earlier version of the manuscript.